# Registration of multi-modal volumetric images of embryos by establishing correspondences between cells

**Abstract.** Early development of an animal from an egg involves a rapid increase in cell number and several cell fate specification events which are accompanied by dynamic morphogenetic changes. In order to correlate the morphological changes with the underlying genetic events, one typically needs to monitor the living system with several imaging modalities offering different spatial and temporal resolution. Live imaging allows monitoring the embryo at a high temporal resolution and observing the morphological changes during the early development. Confocal images of specimens fixed and stained for the expression of certain genes provide high spatially-resolved static snapshots and enable observing the transcription states of an embryo at specific time points during development. The two modalities cannot, by definition, be applied to the same specimen and thus, separately obtained images of different specimens need to be registered. Biologically, the most meaningful way to register the images is by identifying cellular correspondences between these two imaging modalities. In this way, one can bring the two sources of information into a single domain and combine dynamic information on morphogenesis with static gene expression data. The problem of establishing cellular correspondence is non-trivial due to the stochasticity of developmental processes and the non-linear deformation of the specimen during staining protocols. Here we propose a new computational pipeline for identifying cell-to-cell correspondences between images from multiple modalities and for using these correspondences to register 3D images within and across imaging modalities. We demonstrate this pipeline by combining four dimensional time-lapse showing embryogenesis of Spiralian ragworm *Platyneries dumerilii* with three dimensional scans of fixed *Platyneries dumerilii* embryos stained for the expression of a variety of important developmental transcription factors. We compare our approach with methods for aligning point clouds and show that we match the accuracy of these state-of-the-art registration pipelines on synthetic data. We show that our approach outperforms these methods on real biological imaging datasets. In addition, our approach uniquely provides, in addition to the registration, also the non-redundant matching of corresponding, biologically meaningful entities within the registered specimen which is the prerequisite for generating biological insights from the combined datasets. The complete pipeline is available for public use through a Fiji ([20]) plugin.

**Keywords:** Cell Correspondence, Image Registration, *In-Situ* Hybridisation, *Platynereis dumerilii*, Iterative Closest Point, Shape Context

## 1    Introduction

Development of an animal embryo is a highly dynamic process spanning several temporal and spatial scales, and involves a series of dynamic morphogenetic events that are driven by gene regulatory networks encoded by the genome. One of the major challenges in developmental biology is to correlate the morphological changes with the underlying gene activities [17]. Recent advances in fluorescence microscopy, such as light-sheet microscopy ([11], [23]), allows investigating the spatio-temporal dynamics of cells in entire developing organisms and in a time-resolved manner. The three-dimensional time-lapse data produced by light sheet microscopes contain information about positions, trajectories, divisions and deaths of most cells in the embryo during development. However, such data sets typically lack information about gene activities in the living system.

The molecular information is provided by complementary approaches, such as confocal imaging of fixed specimens stained for expression of a certain gene (following the molecular protocols of whole-mount *in-situ* hybridization (ISH)). The three-dimensional images of the fixed and stained embryos contain information about the spatial position of all cells or nuclei and in addition some cells are specifically labelled to indicate the expression of a gene of interest. Images of many such stained specimens showing expression of different genes at a particular stage of development can be readily collected. In order to systematically connect the molecular state of a cell to its fate during embryo morphogenesis, one needs to detect the cells in both live and fixed imaging modalities and identify cell-to-cell correspondences. This is typically achieved by aligning the images. However, the process of chemical fixation during ISH leads to a global and non-linear shrinking of the specimen. Additionally, the embryos are scanned in random orientations, and each specimen is a distinct individual showing stochastic differences in numbers and positions of the cells. This makes the problem of image registration in this context non-trivial.

We reasoned that since our primary objective is to transfer information between the imaging modalities and since cells (or nuclei) are the units of biological interest, it is more important to establish precise correspondences between equivalent cells across the specimen and modalities and that once this is achieved the registration will be obtained implicitly (Figure 1A). We aimed to solve two matching and registration problems. Firstly, intramodal registration, where different fixed embryos stained for different gene expression patterns are registered to one reference specimen (Figure 1B). When successful, the intramodal registration will transfer information about expression of multiple genes derived from distinct staining and imaging experiments to a single reference atlas. Secondly, intermodal registration where individual fixed and stained specimens are registered to an appropriate matching time-point of a time-lapse series of the same animal species imaged live (Figure 1C). When successful, the intermodal reg-

istration will transfer gene expression information from fixed data to the live imaged specimen where it can be propagated along the developmental trajectories of the cells. In both cases, the common denominator are the labelled nuclei and the task is to establish the correspondences between them as precisely as possible.

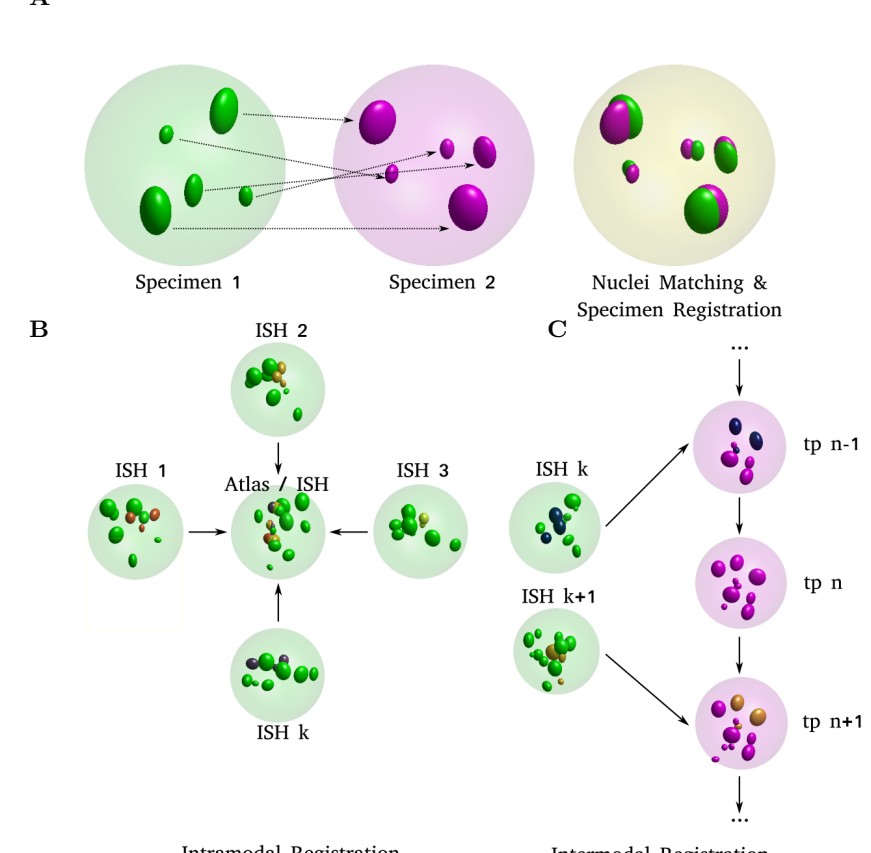

**Fig. 1.** (A) 2-D schematic illustrating the idea: two distinct specimens (left: source and middle: target) are compared in order to estimate pair-wise cell nuclei correspondences and an optimal transform that registers the source onto the target (right) (B, C) 2-D schematic illustrating the two use cases: (left, B) images of distinct, independent *in-situ* specimens, acquired through confocal microscopy are registered to each other, which enables formation of an average, virtual atlas. (right, C) images of *in-situ* specimens, acquired through confocal microscopy are registered to the appropriate frame (tp: time point) in a time-lapse movie acquired through SPIM imaging. Nuclei indicated in darker shades are the ones expressing the transcription factor being investigated. These transcription factor intensities are *transferred* from the source nucleus to the corresponding target nucleus in both use cases.

To address these challenges, we developed a new computational pipeline to identify cell-to-cell correspondences between images from the same and multiple imaging modalities and use them to register the images. We demonstrate the results of the pipeline on fixed ISH images of the embryos of marine annelid worm *Platynereis dumerillii* at 16 hours post fertilization (hpf) and the corresponding long term time-lapse acquired with light sheet microscopy. This worm is particularly suitable for demonstrating our approach because its embryonic development is highly stereotypic, meaning that the number, arrangement and dynamic behaviour of cells is highly similar across individuals. We compare our algorithm with methods for matching point clouds from computer vision such as Coherent Point Drift [18] and a variant of ICP (which we refer to as PCA-ICP) and show that our method outperforms the accuracy of these state-of-the-art global registration pipelines on real biological data. We also perform a series of controlled experiments on synthetic data in order to demonstrate that our method is robust to initial conditions, and noisy nuclei detections. Importantly, the pipeline is made available to the biology research community through an easy-to-use plugin distributed on the Fiji platform [20].

## 2    Related Work

### 2.1    Registration approaches applied to images of *Platynereis dumerilii* embryonic and larval development

*Platynereis dumerillii* has been a playground for image registration approaches over the recent years, due to the efforts to infer gene regulatory networks underlying neuronal development by registering ISH expression patterns. Most of this work has emphasized non-linear registration of an *in-situ* specimen to a virtual atlas. For instance, a new computational protocol was identified to obtain a virtual, high resolution gene expression atlas for the brain sub-regions in embryos at 48 hpf and onwards [22]. The reference signal used in this protocol was the larval axonal scaffold and ciliary bands stained with an acetylated-tubulin antibody. This signal has very distinctive 3D shape within the larva and so this approach relied on intensity based registration where linear transformations were initially applied on the source image to obtain a coarse, global registration. This was then followed by applying a non-linear, deformable transformation which employed mutual-information as the image similarity metric [26].

Another approach, more related to the path we took, leveraged the DAPI image channel (which localises the cell nuclei) to obtain registration of high-quality whole-body scans to a virtual atlas for embryos at stages 48 and 72 hpf [2] and for a larva at 144 hpf [25]. Also these approaches relied on voxel intensities of the DAPI channel rather than on the matching of segmented nuclei as in our approach. Most similar to our work is the approach of [27] where the early lineages of developing embryos were linked to gene expression ISH data by manually identifying corresponding nuclei between embryos imaged in two modalities based on their shape, staining intensity, and relative position.

The embryo specimens targeted in our study are spherical and highly symmetrical, lack distinctive features such as a prominent ciliary band and the nuclei are densely packed. Therefore, intensity based registration approaches using DAPI or neuronal marker channel either fail or perform poorly on such data. Contrary to these approaches and driven by the objective to, first and foremost, transfer gene expression information with cellular resolution between modalities, we adopt a matching-by-detection workflow, where we first detect nuclei in the source and target DAPI image channels and use the detections to estimate an initial transform. We then refine this transform and estimate optimal pair-wise nuclear correspondences. Therefore, after the nuclei detection step, the problem is cast into the realm of point cloud geometric registration methods that has received substantial attention in both biological and computer vision research communities. We discuss the existing approaches in the following two sections.

## 2.2   Matching of cells or nuclei in biological specimens

The work on matching nuclei between biological specimens has focused mainly on *Caenorhabditis elegans* model system that exhibits perfectly stereotypic mode of development, and in fact, every single cell in the animal has its own name. Using this information, a digital atlas was constructed, which labels each nucleus segmentation in a three-dimensional image with an appropriate name. This was initially achieved using a relatively simple RANSAC based matching scheme [15] and was later extended by an active graph matching approach to jointly segment and annotate nuclei of the larva [13]. The *C. elegans* pipelines work well partly due to the highly distinctive overall shape of the larvae and non-homogenous distribution of the nuclei. Another example of matching nuclei between biological specimens is [16] where cell pairings were identified between multiple, independent time-lapse movies showing ascidian development, by identifying a symmetry plane. These publications emphasized nuclei detection and matching between images arising from the same modality. We are not aware of any automated strategy that identified nuclear correspondences between images from different modalities, as we attempt to do (see Figure 1C).

## 2.3   Approaches to Point Cloud Matching in biomedical imaging

In computer vision, a typical workflow for matching point-clouds estimates a rigid or affine transform in order to perform an initial global alignment, which is followed by a local refinement of the initial transform through the Iterative Closest Point (ICP) algorithm. Many global alignment methods identify point-to-point matches based on geometric descriptors [8]. Once candidate correspondences are collected, alignment is estimated from a sparse subset of correspondences and then validated on the entire cloud. This iterative process typically employs variants of RANSAC [6].

One example of geometric descriptors is Shape Contexts, which were introduced by [3] for measuring similarity between two dimensional point clouds and were employed for registering surfaces in biomedical applications ([1], [24]).

These were further extended for use with three dimensional point clouds by [7] and employed for the recognition of three dimensional objects. One characteristic of 3D Shape Context as presented in [7] is that it requires the computation of multiple descriptors for each feature point in the source image. Computing multiple descriptors at a given feature point was deemed excessive and avoided in [21] by identifying a unique local 3D reference frame for each source and target point.

A prominent example of geometric descriptor matching in biological image analysis is the bead-based registration of multiview light sheet (Selective Plane Illumination Microscopy (SPIM)) data [19]. Here, fluorescent beads embedded around a specimen are used as fiduciary markers to achieve registration of 3D scans of the same embryo from multiple imaging angles (referred to as views). This is achieved by building rotation, translation and scale invariant bead descriptors in local bead neighbourhoods, which enables identification of corresponding beads in multiple views and thus allows image registration and fusion of the various views. The approach was extended to multiview registration using nuclei segmented within the specimen instead of beads [12], however the approach is not robust enough to enable registration across different specimen and/or imaging modalities.

A second body of approaches estimate the optimal transform between the source and target point clouds in a single step. One such example is Coherent Point Drift (CPD) algorithm [18] where the alignment of two point clouds is considered as a probability density estimation problem : gaussian mixture model centroids (representing the first point cloud) are fitted to the data (the second point cloud) by maximizing the likelihood. CPD has also been used to perform non-rigid registration of features extracted from biomedical images ([10], [5]). In this paper, we use CPD as one of the baselines to benchmark the performance of our approach.

It is important to note, that in computer vision, matching of interest points represented by geometric descriptors is not the goal but rather the means to register underlying objects or shapes in the images and volumes. Therefore, using a subset of descriptors to achieve the registration is perfectly acceptable and in fact many of the schemes rely on pruning correspondence candidates in the descriptor space to a highly reliable optimal subset. By contrast, in biology, the nuclei that form the basis of the descriptors are at the same time the entities of interest and the goal is to match most, if not all of them ,accurately.

## 3   Our Method

The core of our method is to match the nuclei in the various imaged specimens by means of building the shape context descriptors in a coordinate frame of reference that is unique to each nucleus, this makes the problem of matching rotationally invariant (Figure 2 (II)). The descriptors are then matched in the descriptor space by finding the corresponding closest descriptor in the two specimens and these initial set of correspondences are pruned by RANSAC to achieve

an initial guess of the registration. This alignment is next refined by ICP (Figure 2 (III)). The performance of this part of the pipeline is compared to two baselines, PCA-ICP and CPD (run in affine mode), which are also able to estimate an optimal alignment. At this point, we diverge from the classical approach and evaluate the correspondences through a maximum bipartite matching to achieve the goal of matching every single nucleus from one specimen to a corresponding nucleus in the other (Figure 2 (IV)). The pipeline relies on an efficient nucleus detection method. We present one based on scale-space theory (Figure 2 (I)) but in principle any detection approach can be used as input, to identify feature points to which the Shape Context descriptors would be attached. Also optionally, after the maximum bipartite matching, the estimated correspondence can be used to non-linearly deform the actual images to achieve a registration (Figure 2 (V)). The steps of the pipeline are schematically represented in Figure 2 and are described in detail in the following subsections.

### 3.1   Detecting Nuclei

Following the scale-space theory [14], we assume that the fluorescent cell nuclei visible in the DAPI image channel inherently possess a range of scales or sizes, and that each distinct cell nucleus achieves an extremal response at a scale $\sigma$ proportional to the size of that cell nucleus (Figure 2 (I)). We compute the trace of of the scale-normalized Hessian matrix $H$ of the gaussian-smoothened image $L(x, y, z, \sigma)$, which is equivalent to the convolution ($\circledast$) response resulting from the scale-normalized Laplacian of Gaussian kernel and the image $I(x, y, z)$. The cell nuclei centroid locations (and additional scale information) are then estimated as the local minima of the 4 - D $(x, y, z, \sigma)$ space.

$$\mathrm{trace} H_{\mathrm{norm}} L(x, y, z, \sigma) = \sigma^2 (L_{xx} + L_{yy} + L_{zz})$$

$$L_{xx} = \frac{\partial^2 G_\sigma}{\partial x^2} \circledast I(x, y, z)$$

$$L_{yy} = \frac{\partial^2 G_\sigma}{\partial y^2} \circledast I(x, y, z) \qquad (1)$$

$$L_{zz} = \frac{\partial^2 G_\sigma}{\partial z^2} \circledast I(x, y, z)$$

$$G_\sigma(x, y, z) = \frac{1}{(2\pi\sigma^2)^{\frac{3}{2}}} e^{-\frac{x^2+y^2+z^2}{2\sigma^2}}$$

At this stage, some of the detections might overlap especially in dense, clustered regions. To address this, firstly we employ the assumption that the estimated spherical radius $\hat{r}$ of a cell nucleus is related to its estimated scale $\hat{\sigma}$ through the following relation $\hat{r} = \sqrt{3}\hat{\sigma}$. Next we state a relation drawn from algebra that if $d$ is the distance between two spheres with radii $r_1$ and $r_2$ (and corresponding volumes $V_1$ and $V_2$ respectively), and provided that $d < r_1 + r_2$, the volume of intersection $V_i$ of two spheres is calculated as [4]:

**Fig. 2.** Figure illustrating the key elements of our pipeline: ($I$)  Normalized Laplacian of Gaussian operator is evaluated at multiple $\sigma$. Here, a two dimensional slice of a volumetric image is shown. The operators which provide the strongest local response are shown for three exemplary cell nuclei ($II$) In order to ensure that the shape context geometric descriptor is rotationally covariant, we modify the original coordinate system (shown in gray, top left) to obtain a unique coordinate system (show in black) for each nucleus detection. The Z-axis is defined by the vector joining the center of mass of the point cloud to the point of interest, the X-axis is defined along the projection of the first principal component of the complete point cloud evaluated orthogonal to the Z-axis. The Y-axis is evaluated as a cross product of the first two vectors. Since the sign of the first principal component vector is a numerical accident and thus not repeatable, we use both the possibilities and evaluate two shape context descriptors for each feature point in the source point cloud, in practice. Next, the neighbourhood around each nucleus detection is binned in order to compute the shape context signature for each detection. By comparing shape contexts resulting from the two clouds of cell nuclei detections and following up by RANSAC filtering to prune faulty correspondences, allows us to estimate a global $4 \times 4$-sized affine transform which coarsely registers the source (moving) point cloud to the target (fixed) point cloud ($III$)  In order to obtain a tighter fit between the two clouds of cell nuclei detections, the iterative closest point algorithm is run. The procedure involves the iterative identification of the nearest neighbours (indicated by black arrows), followed by the estimation of the transform parameters ($IV$)  At this stage, a Maximum Bipartite Matching is performed between the transformed moving cloud of cell nuclei detections and the static target cloud of cell nuclei detections, by employing the Hungarian Algorithm for optimization ($V$) Since the two specimens are distinct individuals, non-linear differences would persist despite the preceding, linear registration. We improve the quality of the registration at this stage by employing a thin-plate spline transform and using the correspondences evaluated from the previous step as ground truth control points to estimate the parameters of the thin-plate spline transform

$$V_i = \frac{\pi}{12d} \left(r_1 + r_2 - d\right)^2 \left(d^2 + 2d\left(r_1 + r_2\right) - 3\left(r_1 - r_2\right)\right)^2 \qquad (2)$$

Spheres for which $V_i < \text{t} \times \min\left(V_1, V_2\right)$ are suppressed greedily, by employing a non-maximum suppression step. In our experiments, we use the threshold $t = 0.05$. An optional, additional manual curation of the nuclei detections is made possible through our Fiji plugin.

### 3.2   Finding Corresponding Nuclei between Two Point Clouds

**Estimating a Global Affine Transform** In this section, we will provide the details of our implementation of the 3D shape context geometric descriptor, which is a signature obtained uniquely for all feature points in the source and target point clouds. This descriptor takes as input a point cloud $P$ (which represents the nuclei detections described in the previous section) and a basis point $p$, and captures the regional shape of the scene at $p$ using the distribution of points in a support region surrounding $p$. The support region is discretized into bins, and a histogram is formed by counting the number of point neighbours falling within each bin. As in [3], in order to be more sensitive to nearby points, we use a log-polar coordinate system (Figure 2 (II) bottom). In our experiments, we build a 3D histogram with 5 equally spaced log-radius bins and 6 and 12 equally spaced elevation ($\theta$) and azimuth ($\phi$) bins respectively.

For each basis point $p$, we define a unique right-handed coordinate system: the Z-axis is defined by the vector joining the center of mass of the point cloud to the point of interest, the X-axis is defined along the projection of the first principal component of all point locations in P, evaluated orthogonal to the Z-axis. The Y-axis is evaluated as a cross product of the first two vectors (Figure 2 (II) top). Since the sign of the first principal component vector is a numerical accident and thus not repeatable, we use both possibilities and evaluate two shape context descriptors for each feature point in the source cloud. Building such a unique coordinate system for each feature point ensures that the shape context descriptor is rotationally invariant. Additionally since the chemical fixation introduces shrinking of the embryo volume (the intermodal registration use case, see Figure 1C) and since the embryo volume may considerably differ across a population (intramodal use case, see Figure 1B), an additional normalization of the shape context descriptor is performed to achieve scale invariance. This is done by normalizing all the radial distances between $p$ and its neighbours by the mean distance between all point pairs arising in the point cloud. Similar to [3], we use the $\chi^2$ metric to identify the cost of matching two points $p_i$ and $q_j$ arising from two different point clouds (here $h_i\left(k\right)$ and $h_j\left(k\right)$ denote the K-bin normalized histogram at $p_i$ and $q_j$ respectively).

$$C_{ij} := C\left(p_i, q_j\right) = \frac{1}{2} \sum_{k=1}^{K} \frac{\left(h_i\left(k\right) - h_j\left(k\right)\right)^2}{h_i\left(k\right) + h_j\left(k\right)} \qquad (3)$$

By comparing shape contexts resulting from the two clouds of cell nuclei detections, we obtain an initial temporary set of correspondences. These are

filtered to obtain a set of inlier point correspondences using RANSAC [6]. In our experiments, we specified an affine transform model, sampled 4 pairs of corresponding points without replacement for 20000 trials with an allowed $L_2$ error margin of 15 pixels. We use the Moore-Penrose Pseudo-Inverse operation to estimate the affine transform $A$ between the two sets of corresponding locations.

**Obtaining a tighter fit with ICP** The previous step provides us a good initial alignment. Next, we employ ICP which alternates between establishing correspondences via closest-point lookups (see Figure 2 (III)) and recomputing the optimal transform based on the current set of correspondences. Typically, one employs Horn's approach [9] to estimate strictly-rigid transform parameters. We see equivalently accurate results with iteratively estimating an affine transform, which we compute by employing the Moore-Penrose Pseudo Inverse operation between the current set of correspondences.

**Estimating the complete set of correspondences** We build a $M \times N$-sized cost matrix $C$ where the entry $C_{ij}$ is the euclidean distance between the $i^{\text{th}}$ transformed source cell nucleus detection and the $j^{\text{th}}$ target cell nucleus detection. Next, we employ the Hungarian Algorithm to perform a maximum bipartite matching and estimate correspondences $\hat{X}$ (see Figure 2 (IV)):

$$\hat{X} = \arg \min_X \sum_{i=1}^{M} \sum_{j=1}^{N} C_{ij} X_{ij}, \text{ where } X_{ij} \in \{0, 1\}$$

$$\text{s.t.} \sum_{k=1}^{k=M} X_{ik} \leq 1 \tag{4}$$

$$\sum_{k=1}^{k=N} X_{kj} \leq 1$$

**Estimating a Non-Linear Transform** Since the two specimens being registered are distinct individuals, non-linear differences would persist despite the preceding, linear (affine) registration. We improve the quality of the image registration at this stage by employing a non-linear transform such as the thin-plate spline transform and using the correspondences evaluated from the previous step as ground truth control points to estimate the parameters of the thin-plate spline transform.

## 4   Materials

To test our method, we are using two sets of real biological specimen. Firstly, representing the fixed biological specimen containing information about gene expression, we collected whole-mount specimen of *Platynereis dumerilii* stained with

ISH probes for several different, developmentally regulated transcription factors at the specific developmental stage of 16 hpf. These specimens were scanned in 3D by laser scanning confocal microscopy resulting in three-dimensional images containing the DAPI (nucleus) channel used in our registration as a common reference and the gene expression channel. Secondly, representing the live imaging modality, we obtained access to a light sheet movie capturing the embryological development of the *Platynereis dumerilii* at cellular resolution *in toto* ([23]. The embryos were injected with a fluorescent nuclear tracer prior to imaging and thus the time-lapse visualizes nuclei throughout development. This movie includes the 16 hpf stage of *Platynereis* development providing an appropriate inter-modal target to register the fixed specimen to on the basis of the common nuclear signal.

## 5    Results

We evaluate our proposed strategy on real and simulated data and compare against two competitive baselines. The first baseline, which we refer to as PCA-ICP is an extension of ICP and includes a robust initialization prior to performing ICP. The center of mass of the source point cloud is translated to the location of the center of mass of the target point cloud. Next, the translated source point cloud is rotated about its new center of mass such that its three principal component vectors align with the three principal component vectors of the target point cloud. In order to ensure that the orthogonal system forming the three principal components is not mirrored along any axis, we consider all 8 (+++, ++-, +-+, +--, -++, -+-, --+, ---) possibilities for the obtained principal component vectors of the source point cloud. We initialize ICP from these 8 setups and iteratively estimated a *similar* transform (scale, rotation and translation). Finally, the the configuration which provides the least $L_2$ euclidean distance between the two sets of correspondences, upon the termination of ICP, is kept and the rest of the configurations are discarded.

The second baseline is Coherent Point Drift (CPD) [18]. In our experiments, we executed CPD in the Affine mode with normalization set to 1, maximum iterations equal to 100 and tolerance equal to 1e-10.

We mainly consider two measurements in order to quantify the performance of all considered methods: (*i*) Matching Accuracy which we define as the ratio of the true positive matches and the total number of inlier matches, and (*ii*) Average Registration Error which we define as the average $L_2$ euclidean distance between a set of ground truth landmarks arising from the two point clouds, evaluated after the completion of the registration pipeline. A higher Matching Accuracy and a lower Average Registration Error are desirable readouts to demonstrate better performance.

### 5.1    Experiments on Real Data

For the intramodal registration use case (see Figure 3A & Figure 3C), nuclei detections arising from 11 images of *in-situ* specimens were registered to nuclei

detections arising from the image of a typical, target *in-situ* specimen. Since for real data the true correspondences are not known, we asked expert biologists to manually identify 12 corresponding landmark nuclei. This set represents ground truth landmarks against which we evaluated the results of our registration based on the average $L_2$ euclidean distance of proposed landmark correspondences (Source landmarks are labeled 1 . . . 12 and Target Landmarks are similarly labeled 1' . . . 12' in Figure 3).

For the intermodal registration use case (see Figure 3B & Figure 3D), nuclei detections arising from 7 confocal images of *in-situ* specimens are registered to the corresponding frame from the time lapse movie which contains an equivalent number of nuclei. They were similarly evaluated on the average $L_2$ euclidean distance in the positions of landmarks identified in the movies by the expert annotators.

The results show that after applying our proposed pipeline, the average registration error of corresponding landmarks is around 25 and 35 pixels for intramodal and intermodal registration use cases respectively (Figure 3E). The accuracy is significantly better compared to the baseline methods. The exemplary intramodal image shows good overlap of the nuclear intensities (Figure 3C). The displacement of the corresponding landmarks (denoted by the yellow unprimed numbers) is better in the left part of the specimen compared to the right part. This suggests that significant non-linear deformation occurred during the staining process and our current pipeline relying on affine models is unable to undo this deformation. For the intermodal registration, the pipeline clearly compensated for the mismatch in scale between the fixed and live specimen (Figure 3D). The remaining error is, similarly to the intramodal case, likely due to non-linear distortions. In terms of matching accuracy after performing maximum bipartite matching, our method outperforms the baselines (they are however not optimised for this task). Since the matching accuracy is estimated on only 12 corresponding landmarks, which represents only 3.6 % of the total matched nuclei, it is likely subject to sampling error. This is reflected by the broad spread of accuracy for both inter- and intramodal use cases (Figure 3F).

Since obtaining a larger sground truth correspondences for a larger sample is not practical we turn next to evaluating the approach on synthetic data.

## 5.2   Experiments on Simulated Data

Starting from the nuclei detection on real fixed embryos, we generated simulated ground truth data by random translation, rotation and scaling operations, followed up by (*i*) adding gaussian noise to the location of individual segments (i.e. nuclei) and (*ii*) randomly adding nuclei (Figure 4A). The simulated embryos are meant to resemble the live-imaged embryos which in real scenarios are also rotated, translated and scaled compared to the fixed specimens and may have extraneous or missing nuclei due to biological variability or segmentation errors.

**Robustness to Gaussian Noise** The synthetic 'live embryos' were generated by manipulating nuclei detections from multiple, independent *in-situ* specimens.

First, the nuclei detections of each *in-situ* specimen are provided a random translation offset, next the translated point cloud is rotated by a random angle between -30 deg and +30 deg about an arbitrary axis passing through the center of mass of the point cloud, and finally, the translated and rotated point cloud is scaled by a random factor (See Figure 4A). We add five levels of Gaussian noise with standard deviations 0, 5, 10, 15 and 20. For a given standard deviation, gaussian noise is independently added to the x, y, and z-axes of each transformed nucleus detection (point).

The results of evaluation of matching accuracy with respect to different levels of Gaussian noise show that all methods provide equivalent performance (Figure 4B). The matching accuracy starts to break down when the magnitude of gaussian noise > 10 pixels.

**Robustness to Outliers** In order to test robustness against over or under-segmentation of nuclei, we add outliers to both the source fixed *in-situ* volumes and the corresponding simulated 'live embryo'. New outlier points are generated by sampling existing points and adding a new point at a standard deviation of 20 pixels from their locations. The results show that the CPD Affine method performs the best in the presence of outliers, while our approach is more stable compared to the PCA-ICP (Figure 4C).

## 6   Discussion

Our method showed promising results on real biological data in terms of registration accuracy and provided equivalent performance when compared to state of the art methods on simulated data. The pipeline offers several avenues for further improvement towards achieving more precise one-to-one matching of cells within and across imaging modalities for separate biological specimen. One area open for future investigations is certainly obtaining accurate segmentations which delineate the shape of the nuclei.

Another improvement may come from the definition of the 3D geometric descriptor. Our implementation of shape context as a 3D geometric descriptor draws from [3]. Here, we use a log-polar coordinate system and build 3D histograms by evenly dividing the azimuth and elevation axis. This creates two issues $(i)$ bins of varying sizes are obtained which need to be compensated for and $(ii)$ degenerate bins near the pole are obtained, which makes the matching of feature points non robust to noisy detections. These drawbacks could be addressed through two approaches: $(i)$ employing the optimal transport distance between two 3D histograms would provide a more natural way of comparing two histograms as opposed to the current $\chi^2$ squared distance formulation, $(ii)$ opting for a more uniform binning scheme (see for example, [28]) would eliminate the issue of noisy detections jumping arbitrarily between bins near the poles. Finally, the method will benefit from non-linear refinement as the specimens are often deformed in an unpredictable manner during the staining and imaging protocols.

By establishing nuclei correspondences between images of *in-situ* specimens and the time lapse movie, biologists will be able transfer the gene expression information from the fixed specimens to the dynamic a cell lineage tree generated by performing lineage tracing on the time-lapse movie. This will enable biologists to study the molecular underpinning of dynamic morphogenetic processes during embryo development.

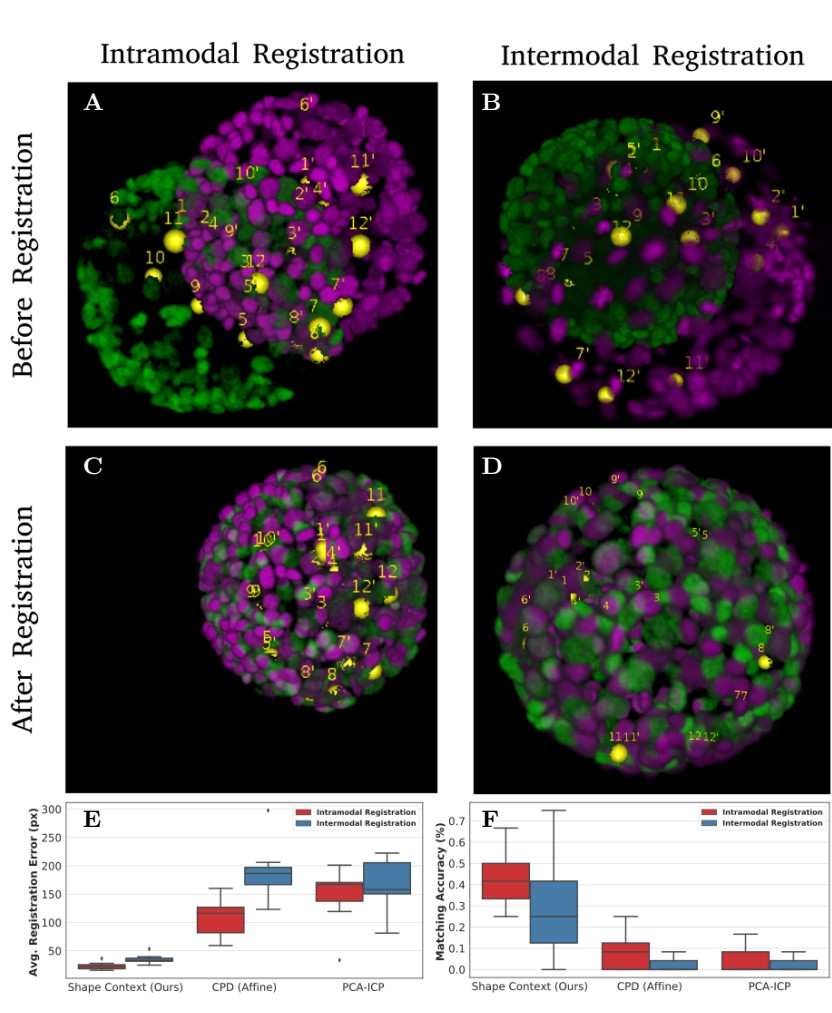

**Fig. 3.** A: DAPI channels indicating cell nuclei for two distinct *in-situ* specimens (source: green and magenta: target image). B: DAPI channels indicating cell nuclei for an *in-situ* specimen (source: green) being registered to the corresponding frame containing equivalent number of cell nuclei, in the time-lapse movie (target: magenta). Landmarks for source image are indicated as yellow spheres and labeled from 1 . . . 12. Similarly, landmarks for the target image are labeled from 1' . . . 12'. C, D: After performing registration with our proposed pipeline, corresponding landmarks from the source and target image appear to be much closer in Euclidean distance. E: Plot indicating the average Euclidean distance between landmarks after applying different registration pipelines. F: Plot indicating the percent of correct correspondences between landmarks, evaluated through Maximum Bipartite Matching, after applying different registration pipelines.

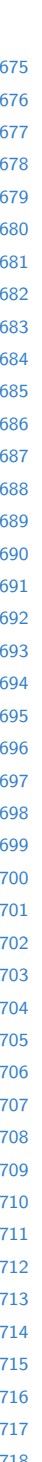
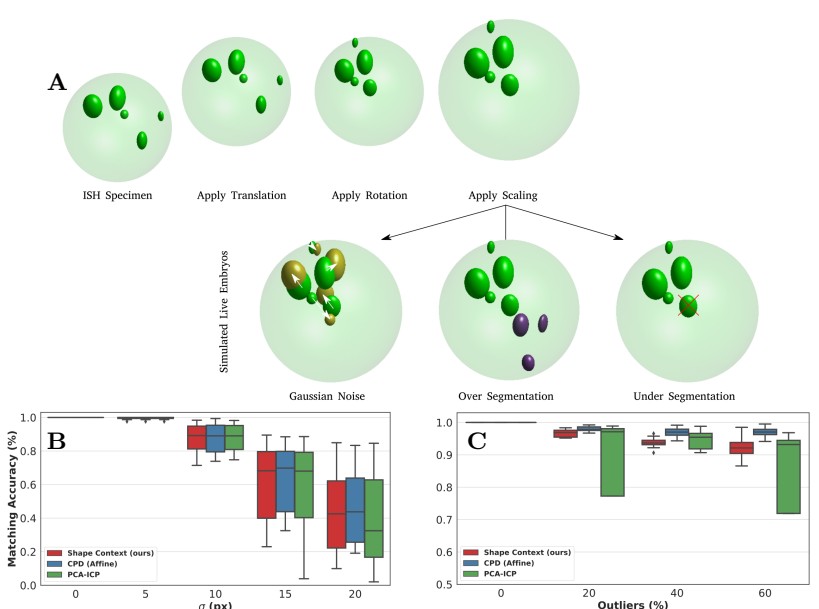

**Fig. 4.** Live embryos are simulated by manipulating cell nuclei detections from multiple *in-situ* specimens. A: First, the cell nuclei detections of each *in-situ* specimen are provided a random uniform translation offset, next the translated point cloud is randomly rotated by an angle $\in \{-\pi/6, \pi/6\}$ about a random axis passing through the center of mass of the translated point cloud, and finally, the translated and rotated point cloud is scaled by a random factor. B: We add five levels of Gaussian noise with standard deviations 0, 5, 10, 15 and 20 to explore robustness to gaussian noise. For a given standard deviation, Gaussian noise is independently added to the x, y, and z-axes of each transformed nucleus detection (point). C: We add outliers to both the *in-situ* and the corresponding simulated live embryo. New outlier points are generated by sampling existing points and adding a new point at a standard deviation of 20 pixels from their locations

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
