# OpenReview forum: "Registration of multi-modal volumetric images of embryos by establishing correspondences between cells"
_thecvf.com/ECCV/2020/Workshop/BIC — BIC 2020 Oral_

### Official Review · AnonReviewer2 · 2020-07-30
**Excellent work**

**Rating:** 10
**Confidence:** 4

**Review:**

The paper presents a methodology for registration of high-resolution gene expression data and live microscopy images of Platynereis dumerilii embrios in 3D. The method detects nuclei, computes descriptors, finds correspondences between cells and aligns the data. One of the core strategies used in this methodology is the ability to match single cells between two image sets, and by solving this, the registration is recovered accurately and efficiently.

The paper is very well written and tackles an important and challenging biological problem. The proposed methods are innovative and creative, and have shown to be effective for solving the task with the best precision possible. The evaluation is conducted in synthetic data as well as in real world images and the results are very robust. Clean study and elegant solution.

**Reviews Visibility:**

I agree that my anonymized review is made publicly visible, if the submission is accepted.

---

### Official Review · AnonReviewer1 · 2020-07-30
**A difficult and important registration problem is solved by a creative combination of different algorithms.**

**Rating:** 10
**Confidence:** 4

**Review:**

The authors address the problem of cell-to-cell registration of the stereotypic specimen acquired by different imaging modalities. This is an important problem in developmental biology which has not yet received sufficient attention from the computer vision community. The task can be divided into two registration problems: registration between static samples and between static and time-lapse imaging. Shape context descriptors are used to represent different nuclei. The proposed method is compared to strong baselines, outperforming them on real and simulated data. The method is described in detail, including the outlook on potential future improvements.

**Reviews Visibility:**

I agree that my anonymized review is made publicly visible, if the submission is accepted.

---

### Decision · Program_Chairs · 2020-07-31

Accept (Oral)